# Comparison of Flavor Profile Relationship of Soy Sauce under Different Storage Conditions

**DOI:** 10.3390/foods12142707

**Published:** 2023-07-14

**Authors:** Rui Wang, Miao Liang, Zhimin Zhang, Yajian Wu, Yuping Liu

**Affiliations:** 1China Food Flavor and Nutrition Health Innovation Center, Beijing Technology and Business University, Beijing 100048, China; wangruicoke@163.com (R.W.); liang_miao923@163.com (M.L.); ss1765581@163.com (Z.Z.); 2130031020@st.btbu.edu.cn (Y.W.); 2School of Light Industry, Beijing Technology & Business University, Beijing 100048, China

**Keywords:** soy sauce, storage, accelerated aging, flavor, key odorants, odor activity value

## Abstract

To elucidate the relation of flavor in soy sauce (SS) kept at room temperature (SSAT) and SS kept under accelerated aging condition (SSAA), four analytical instruments, including electronic nose (E-nose), electronic tongue (E-tongue), gas chromatography–mass spectrometry-olfactory combined with solvent assisted flavor evaporation, and amino acid analyzer, were applied for analyzing the overall flavor profiles and flavor constituents in SSAT and SSAA. The results of E-nose and E-tongue showed overall flavor profile in SSAT for 3 weeks was similar to that of SSAA for 1 week, and 6 weeks (SSAT) was similar to 2 weeks (SSAA). In SS, a total of 35 odor-active compounds were identified and quantitated, and 22 compounds with odor activity value ≥1 were determined as key odorants. The compounds with the highest concentration were 4-hydroxy-2,5-dimethyl-3(2H)-furanone (28,756 μg/mL), followed by acetic acid (8838 μg/mL) and maltol (7984 μg/mL). The heatmap and hierarchical cluster analysis indicated that the concentrations of key odorants and amino acids in SSAT for 3 weeks was close to those of SSAA for 1 week, and 6 weeks (SSAT) was similar to 2 weeks (SSAA). Based on the results obtained above, it was concluded that the flavor changes in SSAA for 1 week were equivalent to those in SSAT for 3 weeks.

## 1. Introduction

Soy sauce (SS) is a traditional fermented condiment in Southeast Asia, which is widely used in daily life, and its consumption is expanding worldwide due to its unique flavor [1]. SS is mainly made from soybeans, soybean meal, wheat flour, and wheat bran through microbial fermentation [2]. Firstly, the materials were cooked, then mixed and inoculated with microbes (such as *Aspergillus oryzae*, *Aspergillus niger*, etc.); then, they were fermented to obtain koji, which was immersed in a brine solution with a certain concentration to form moromi; and finally, moromi was further fermented for some time, filtered, and pasteurized to obtain SS. During moromi fermentation, some other microbes (such as yeast, lactic acid bacteria, etc.) also participated in the fermentation process to decompose materials into small molecules. Therefore, SS contains a variety of amino acids, peptides, vitamins, and aroma compounds that contribute to its characteristic flavor [3,4]. Flavor is the most important indicator in determining the quality and consumer acceptance of SS, and it is related to the presence of volatile and non-volatile components, which should be investigated together [5,6]. SS is a complex matrix, and its overall flavor profile will change during storage. Accelerated aging test is often used to investigate the odor and taste changes in food during storage. However, there are few publications examining the similarities and differences of overall flavor profile of SS kept at room temperature (SSAT) and under accelerated aging (SSAA) condition [7].

The use of combining electronic nose (E-nose) with electronic tongue (E-tongue) allows for a precise evaluation of odor and taste information of the sample investigated. At present, the combination of E-nose and E-tongue has been widely used to distinguish dairy products, tea, duck meat products, and other food [8,9]. According to the results of Guan et al., principal component analysis (PCA) and radar plots of E-nose and E-tongue exhibited how the flavor profiles of flour baked more than 8 min were significantly different from the original flavor of unbaked flour [10]. E-nose could effectively distinct the odors of different cultivars apples, and E-tongue analysis results showed that the sourness, saltiness, and umami were different among these apples [11].

Aroma-active compounds contribute to the odor of SS. Wang et al. found that different heating conditions could change the intensities of caramel, nut, and spiciness in SS during cooking through the changing concentrations of corresponding aroma compounds [12]. Liang et al. investigated the differences in flavor characteristics between raw SS and heated SS, and found that spicy, caramel, and fruity attributes had higher intensities in heated SS compared to raw SS [13]. Normally, amino acids in SS have greater contributions to the flavor, and their concentrations are one of the important factors for evaluating SS quality. Zhou et al. found that added amino acid resulted in more produced malty flavors, higher recognition of floral and fruity notes, a 43.26% increase in 4-ethylphenol (smoky-like), and a 42.83% increase in ester concentration compared to the control [14]. There are no reports on the combined use of E-nose, E-tongue, gas chromatography–mass spectrometry-olfactory (GC-MS-O), and amino acid analyzer in SS.

The aims of this present study were: (1) to compare the overall flavor profile of SS kept at room temperature (SSAT) and SS kept under accelerated aging condition (SSAA) by means of E-nose and E-tongue; (2) to determine the key odorants in SSAT and SSAA; (3) to measure the concentrations of the amino acids in SSAT and SSAA; (4) to find the samples with the similar flavor profile between SSAT and SSAA; and (5) to elucidate the relation between SSAT and SSAA. This study is helpful to investigate in detail the flavor changes in SS during storage.

## 2. Materials and Methods

### 2.1. Soy Sauce Samples

Samples of the same production batch of Jiajia Brewing SS produced in November 2020 were purchased from a Beijing local supermarket in October 2021. The raw materials of SS samples were water, non-transgenic defatted soybeans, wheat, edible salt, sugar, sodium glutamate, disodium 5′-ribonucleotide, yeast extract, and sucralose. A1–A8 are SS stored at room temperature (SSAT) at 24 ± 1 °C for 1–8 weeks, and AH1 and AH2 are SS stored under accelerated aging condition (SSAA) at 37 ± 1 °C for 1 week and 2 weeks, respectively. All samples were stored in a 4 °C refrigerator before analysis.

### 2.2. Chemicals

Furfuryl alcohol (98%), 1-butanol (99.5%), methionol (98%), 3-methyl-1-butanol (99%), maltol (99%), methyl cyclopentenolone (99%), γ-butyrolactone (99%), ethyl 4-hydroxy-3-methoxybenzoate (98%), guaiacol (99%), 4-ethyl-2-methoxyphenol (98%), propionic acid (99%), 4-methylpentanoic acid (99%), isobutyric acid (99%), 3-methylbutanoic acid (99%), methylpyrazine (98%), 2,3,5-trimethylpyrazine (98%), 2,6-dimethylpyrazine (98%), isobutanol (99%), 2-octanol (98%), 2-isopropylphenol (98%), 2-methyl-3-heptanone were obtained from J&K Chemicals Ltd. (Beijing, China); phenylacetaldehyde (95%), methional (98%), 1-hydroxy-2-butanone (98%), 2(5H)-furanone (98%) were purchased from Macklin Biochemical Co., Ltd. (Shanghai, China); 4-hydroxy-2,5-dimethylfuran-3(2H)-one (98%) were purchased from Aladdin Reagents Co., Ltd. (Shanghai, China); acetol (>80%), 4-ethylphenol (>97%), 3-methyl-2(5H)-furanone (>97%) were bought from TCI Chemical Ltd. (Shanghai, China) acetoin (97%), 2-ethyl-5-methylpyrazine (>98%) were purchased from Adamas Reagents Co., Ltd. (Shanghai, China); n-alkanes (C6~C30) were obtained from Aldrich Chemical Co., Ltd. (Shanghai, China); both dichloromethane and sodium sulfate anhydrous (analytical grade) were obtained from Sinopharm Chemical Reagent Co., Ltd. 5-ethyl-4-hydroxy-2-methyl-3(2H)-furanone (97%) were obtained from Ark Pharma Scientific, Ltd. (Chicago, IL, USA). Other chemical reagents were analytical grade, which were obtained from Sinopharm Chemical Reagent Co., Ltd. (Beijing, China) and Sykam Co., Ltd. (Beijing, China), respectively.

### 2.3. Electronic Nose Measurement

The electronic nose experiment was conducted based on a previous study with slightly modified [15]. The odor characteristics of SS were analyzed using a portable PEN3 E-nose device (Win Muster Airsense Analytics Inc., Schwerin, Germany). The PEN3 E-nose consists of an array of 10 different metal oxide sensors, a gas flow control system, and analysis control software. The metal oxide semiconductors are sensitive to their corresponding odorants. The sensor performance is listed in Table 1. The odor profiles of 10 SS samples (0-week sample was used as control group) were measured, and 0.5 mL SS samples were accurately weighed and placed in a 10 mL headspace bottle with an airtight spacer. The equipment parameters were as follows: measurement time of 120 s, cleaning time between the same sample for 60 s, cleaning time between different samples for 120 s, injection flow of 300 mL/min. Each analysis was repeated 3 times for SS sample. Data processing was performed using Win-muster E-nose software.

### 2.4. Electronic Tongue Measurement

The electronic tongue experiment was carried out on a previous study with slightly modified [16]. The E-tongue simulates the human taste system for evaluating food taste, using artificial lipid membrane sensor technology so as to detect taste indicators in food. Taste analysis was applied with SA402B system (Intelligent Sensor Technology Co. Ltd., Tokyo, Japan), which had sensitive sensors that could respond to different tastes. The six chemical sensors were: CA0 for sourness, C00 for bitterness and aftertaste bitterness, AE1 for astringency and aftertaste astringency, CT0 for saltiness, GL1 for sweetness, and AAE for umami and richness. Before analysis, sensors were pretreated in a reference solution (30 mmol/L potassium chloride solution containing 0.3 mmol/L tartaric acid). Sweetness was detected separately from the other tastes. The electrical potential of the taste sensor film was detected based on the phase change with the reference electrode. Before measurement, the E-tongue is subjected to self-test, activation, calibration, and diagnostic steps to ensure the reliability and stability of the collected data. For the analysis, SS sample diluted 30 times with odorless deionized water was placed in a special beaker for the E-tongue analysis. Basic tastes were measured for four times, and the sweetness was measured for five times. The last three measurements were taken for statistical analysis.

### 2.5. The Isolation of Volatiles by SE-SAFE

The isolation of volatiles in SS was performed based on the previous study with some modifications [17]. SS sample (70 mL) was mixed with 50 mL dichloromethane. The mixture was shaken for 1 h at 180 rmp (revolutions per minute) in a thermostatic oscillator (ZWY-100H, Shanghai Zhicheng Analytical Instrument Manufacturing Co., Ltd., Shanghai, China), and then, was centrifuged at 4 °C, 8000 rmp in a centrifugal (H1750R, Hunan Xiangyi Instrument Manufacturing Co., Ltd., Hunan, China). The organic phase and aqueous phase were separated. Afterwards, the above operation was repeated two times. The organic phases obtained above were combined together to obtain SS extract. The extract went through a SAFE device at 40 °C under a vacuum of 2 × 10^−5^ mbar (Edwards TIC Punping Station from BOC Edwards, English) to separate the volatiles from the non-volatiles. The distillate obtained was dried with anhydrous sodium sulfate overnight, and then, further concentrated to 5 mL using a Vigreux column (50 cm × 1 cm) (Beijing Jingxing Glassware Co., Ltd., Beijing, China) at 48 °C, and finally, concentrated under a gentle nitrogen stream to approximately 500 μL, so as to perform gas chromatography–mass spectrometry-olfactory (GC-MS-O) analysis. The isolation experiment was conducted three times.

### 2.6. GC-MS-O Analysis Conditions

The volatiles obtained above were analyzed by gas chromatography (Agilent model 7890B) equipped with an Agilent 5977A mass spectrometer detector (MSD) and an olfactory detector port (OPD-3; Gerstel, Mulheim an der Ruhr, Germany). All SS volatile extracts were isolated on both DB-WAX and HP-5MS capillary column (30 m × 0.25 mm × 0.25 μm; Agilent Technologies, Santa Clara, California, USA) with helium (≥99.999% purity) at a flow rate of 1.7 mL/min as the carrier gas, respectively. The injected volume was 1 μL, and spitless mode was used. The heating program was as follows: the initial oven temperature was 40 °C which was maintained for 2 min, then increased to 80 °C at a rate of 8 °C/min, and then, increased to 100 °C at a rate of 4 °C/min, and finally, raising at a rate of 6 °C/min to 230 °C, and holding 15 min. The temperature of the injection port was set at 240 °C, while the ion source temperature was set at 230 °C and the transfer line between GC and MS was set at 250 °C. Detection was carried out in electron ionization (70 eV) with full scan mode, and the scanning range (*m*/*z*) from 30 to 350. The sniffing port was equipped with humidified air to maintain the nose sensitivity, and the temperature of the olfactory was 120 °C. GC-MS-O analyses were performed by three trained sensory panelists (1 female and 2 males, from Beijing Technology and Business University), and each panelist were sniffed 3 times at least.

### 2.7. Qualitative Analysis of Volatile Compounds

Odor-active compounds detected in SS were identified by comparison of their mass spectra data, retention indices (RI), and odor characteristics with those of authentic compounds. The RI of each compound was calculated by using the retention time (RT) of the compound and the RTs of n-alkanes adjacent to the compounds.

### 2.8. Quantitative Analysis of Odor-Active Compounds

Odor-active compounds in SS were quantitated by using 2-methyl-3-heptanone, 2-octanol, and 2-isopropylphenol as internal standards. The three internal standards (300 μL, 10^−4^ g/mL in dichloromethane) were added into SS samples before extraction experiment; the volatile extract was obtained based on the method used above. The extract obtained was measured by GC-MS under the same conditions used above except selected ion monitoring (SIM) mode used. The quantitative analysis experiment was performed three times; the result was the average value of three experiments. The correction factor for each odor-active compound were measured by analyzing the authentic compound and internal standards with the same mass; its value was calculated according to the peak areas of the authentic compound and internal standards.

### 2.9. Amino Acid Determination

The determination was employed on a previous study with slightly modified [18]. Referring to the National Food Safety Standard (GB 5009.124-2016), the amino acid compositions of SS were determined by SYKAM 433 amino acid analyzer (Sykam, Germany) using post-column derivatization with ninhydrin. Analytical column (4.6 mm × 150 mm, Sykam, Germany) was used for amino acid determination; column temperature was 58 °C. Mobile phase was sodium citrate buffer (pH = 3.4) and reaction solution (Ninydrin solution, Sykam) with the flow rate 0.45 mL/min and 0.25 mL/min, respectively. SS samples were hydrolyzed with hydrochloric acid (6 M) at 110 °C for 22 h. Among the 16 amino acids, proline was measured at a wavelength of 440 nm, and the remaining 15 amino acids were measured at 570 nm.

### 2.10. Data Analysis

Quantitative data were reported as mean value ± standard deviation (SD). All tables were drawn using Microsoft Office Excel 2021. One-way analysis of variance (ANOVA), Duncan’s multiple-range tests and hierarchical cluster analysis (HCA) were performed by using the IBM SPSS Statistics 26 software; a difference of *p* < 0.05 was considered as significant. Heatmap were performed by TB tool; origin 2021b (Origin-Lab, Northampton, MA, USA) and SIMCA14.1 was used to draw the radar chart and PCA, respectively.

## 3. Results and Discussion

### 3.1. Electronic Nose Analysis

E-nose can be used to obtain complete information related to volatile compounds in a sample and are a better way to analyze the overall odor profile. Figure 1 showed the PCA of the E-nose sensors response to the volatiles in SS samples. The total variance contributed by the principal components (PC) in Figure 1 were 85.4%, demonstrating that PC1 and PC2 reflected most of the information about the overall odor profile of the SS samples. AH1 and A3 were similar in distance, indicating that they had similar odor characteristics. While AH1 was farther away from A1 and A2, they were therefore significantly different and were distinguished easily. Obviously, there was a clear overlap among AH2 and A7, which showed that both AH2 and A7 had a similar overall odor profile. Additionally, AH2 and A6 were also similar in distance. From the results obtained above, it could be seen that the overall flavor profile of A3 (SSAT) was similar to that of AH1 (SSAA); A6 (SSAT) and A7 (SSAT) were similar to AH2 (SSAA).

### 3.2. Electronic Tongue Analysis

E-tongue converts electrical signals into taste signals that are used to discriminate the taste of foods, and it can eliminate the subjective errors from the panelists during sensory evaluation. Figure 2 presented taste radar chart of SS with different storage conditions, where SS stored for 0-week was set as a control group. In Figure 2a (A1–A4 and AH1), the taste profile of AH1 was similar to that of A3, which was consistent with the E-nose results. Furthermore, A2 and A4 had sweeter tastes. The sweetness of SS was associated with small molecule sugars which were derived from the degradation of starchy chains from raw materials during fermentation process. Meanwhile, sweet amino acids produced by protein degradation such as glycine, alanine, threonine and some small, sweet peptides also contributed to the sweetness of SS [19]. In Figure 2b (A5–A8 and AH2), the overall taste profile of AH2 was closer to that of A6, which meant both had a similar taste. Among them, AH2 possessed a stronger sour taste which was related to the sour taste compounds contained, such as acetic acid, propionic acid, etc. A5 had a stronger bitter taste originating from bitter amino acids, such as phenylalanine, arginine, tyrosine, and bitter peptides produced by excessive hydrolysis of protein from raw materials [20]; this result was consistent with the analysis results of amino acid in SS.

PCA is a statistical analysis method that reduces multiple indicators to less indicators which reflect as much information about the original variables as possible. When the cumulative variance contribution was greater than 80%, the results of PCA were considered as reflecting the major information of the sample. Figure 2c showed the PCA results of 10 SS samples at different storage conditions. The variance contributions of PCA1 and PCA2 were 53.4% and 32.8%, respectively. The cumulative contribution values were able to reflect 86.2% of the information of the original taste. From Figure 2c, it could be seen that the distance of AH1 was close to that of A3 and A4, and AH2 close to that of A6 and A7. That is, the taste characteristics of AH1 (SSAA) were similar to those of A3 and A4 (SSAT), and AH2 (SSAA) was similar to A6 and A7 (SSAT). The results obtained were nearly in line with that of E-nose analysis.

### 3.3. Analysis of Odor-Active Compounds

A total of 35 odor-active compounds were sniffed and identified in 10 SS samples. Table 2 listed all the compounds; Table 3 showed their concentrations. These odorants concluded seven alcohols, five ketones, six acids, three phenols, two aldehydes, five pyrazines, three esters, and four furans, and they had been found in previous reports as odor-active compounds in SS [21,22,23].

#### 3.3.1. Alcohols

Alcohols were mainly produced by the metabolism of sugars and amino acids under aerobic conditions, and some alcohols were also formed by yeast through the conversion of related aldehydes [24]. In total, 7 alcohols were identified in the 10 samples; methionol (6423–9689 μg/L) and 3-methyl-1-butanol (5677–8163 μg/L) probably contributed greatly to the overall odor profile of SS because of their high concentrations. Among seven alcohols, methionol—with the highest concentrations—was a sulfur-containing compound, and was described as having a cooked potato-like odor. Phenylethyl alcohol (4226–5776 μg/L) with a floral aroma was also a significant odorant in SS, and had been identified as an important odor-active compound in Korean fermented SS [25]; it could be formed by degradation of phenylalanine during the fermentation of sauce mash. Additionally, maltose and lactose as well as other substances could react with amino acids to form furfural which changed into furfuryl alcohol during storage process, which made the concentration of furfuryl alcohol also increase. However, furfuryl alcohol was supposed to be dreadful to the overall flavor of SS. ANOVA showed that the concentration of isobutanol, acetol, furfuryl alcohol, and methionol were similar in A6 and AH2.

#### 3.3.2. Furans

Furanones were considered to be the vital odor compounds in foods, especially in SS. Four furans were identified in this study, including 4-Hydroxy-2(or 5)-methyl-5(or 2)-ethyl-3(2*H*)-furanone (HEMF), 4-Hydroxy-2,5-dimethylfuran-3(2*H*)-one (HDMF), 2(5H)-furanone, and 3-methyl-2(5H)-furanone. HEMF and HDMF imparted SS caramel odor, and had been determined as the key odorants in Japanese SS [26,27]; they could be produced by certain intermediates of the Maillard reaction of pentose during the heating process [28]. Among all of the odor-active compounds identified in this study, the content of HEMF (25,929–39,729 μg/L) was the most abundant, accounting for 80% of furans. The generation of HEMF was influenced by various factors, such as fermentation temperature, time and strains of microbiology, etc. HEMF was stable in SS, and had a strong aroma like a sweet Western dessert with very low sensory threshold (22.3 μg/kg) [29]. Furthermore, Japanese scholars had found that HEMF also had anti-tumor and anti-cancer effects. Additionally, HDMF was also considered as the important odor-active compound in Chinese SS [30] and Japanese SS [26], which was with the concentrations of 3380–5689 μg/L in this present study. Moreover, concentration of both compounds in AH1 was significantly higher than A1–A4 samples, which implied that heating could increase the contents of them. ANOVA showed that changes of the concentrations of 2(5H)-furanone (546–620 μg/L), 3-methyl-2(5H)-furanone (236–255 μg/L) in A3, A4, and AH1 samples were very little.

#### 3.3.3. Acids

In total, 6 acids, including acetic acid, propionic acid, isobutyric acid, 3-methylbutanoic acid, 4-methylpentanoic acid, and phenylacetic acid, were identified in the 10 SS samples. These acids might be formed by two pathways. One was from the deamination of amino acids [21], and the other came from degradation of fatty acids in soybean under the lipases produced by *Aspergillus oryzae* and *Aspergillus niger* [31]. These acids could impart SS sour, cheesy, sweaty, and honey odor. Among all identified acids, acetic acid had the highest concentration (8813–12,284 μg/L), accounting for about 55% of total concentration of acid compounds. Except acetic acid, propionic acid and phenylacetic acid also had higher contents; their concentrations were 2249–3071 μg/L and 2167–4518 μg/L, respectively. 3-Methylbutanoic acid (860–1155 μg/L) was also reported as the main aroma compound in Korean soy sauce and barley bran sauce [31,32]. Propionic acid and 3-methylbutanoic acid had similar concentration in A3 and AH1; acetic acid and 4-methylpentanoic acid had similar content in A6 and AH2. Furthermore, the total levels of acids in AH2 (21,391 μg/L) were higher, which indicated heating even at lower temperature was beneficial to the formation of organic acids. This were consistent with the E-tongue analysis results.

#### 3.3.4. Ketones

Five ketones were detected, and they were mainly from raw materials used and fermentation process. Maltol (sweet) was the most abundant ketone with a concentration of 7984–10,597 μg/L, followed by 1-hydroxy-2-butanone (4657–6304 μg/L). It was worth noting that the RTs of 2-acetyl-1-pyrrolidine (2AP) and ethyl lactate were very close; the concentration of ethyl lactate (2736–3997 μg/L) in SS was so much more than that of 2AP (40.4–96.0 μg/L) that the peak of 2AP was often covered by that of ethyl lactate, so 2AP was seldom identified in SS by GC-MS. However, because the threshold of 2AP (0.12 μg/kg) was much lower than that of ethyl lactate (50,000 μg/kg), 2AP could be sniffed easily by GC-MS-O. 2AP was formed by Maillard reaction during the heating process of food [33], and it was detected in bread crusts and wheat. Previous studies had demonstrated that ornithine was a precursor of 2AP [34]. Moreover, valine can produce chocolate aroma; histidine, lysine, and proline can produce bakery aroma, and these amino acids might be the precursor of some sweet aroma compounds, such as acetoin and maltol. These amino acids and sweet aroma components took on the similar change trend during SS storage. 1-Hydroxy-2-butanone and 2AP had similar content in A3 and AH1; acetoin had similar content in A6 and AH2.

#### 3.3.5. Phenols

Three phenols sniffed were guaiacol, 4-ethyl-2-methoxyphenol (4EG), and 4-ethylphenol, and they had been identified as odor-active compounds in SS and gave SS smoky odor. Both 4EG and 4-ethylphenol were produced from the degradation of lignin glycosides in cereal bran during yeast fermentation [24]. Guaiacol and 4EG with smoky odor had been reported as key odorants in Japanese SS [27]. Among phenols, 4EG was predominant (4512–5892 μg/L), and accounted for 90% of the total phenolics content. It was reported that the generation of phenolics could be affected by SS production conditions; for example, the odor contribution of phenolic compounds increases significantly after pasteurization of raw SS [22].

#### 3.3.6. Esters

Esters had fruity odor; three esters, including ethyl lactate, ethyl 4-hydroxy-3-methoxybenzoate (ethyl vanillate), and *γ*-butyrolactone, were identified in this present study. Ethyl lactate and *γ*-butyrolactone were reported as odor-active compounds in Chinese SS [19,30]. These esters were primarily formed via the esterification reaction of alcohols with fatty acids during the production of SS [24]. Among three esters, the concentration of ethyl lactate (2736–3997 μg/L) was the highest in all samples, which were described as having a fruity aroma. The content of *γ*-butyrolactone and ethyl lactate were similar in A3 and AH1. Furthermore, ethyl vanillate was rarely reported in SS, and it was detected at a concentration of 77–194 μg/L in this study. However, it was often identified in wine and sometimes in the berries [35].

#### 3.3.7. Pyrazines

As odor-active compound, five pyrazines were sniffed. Pyrazines were nitrogen-containing heterocyclic compounds with strong sensory properties in foods, and they were presented in mostly cooked meats [36]. Most pyrazines were produced by self-condensation and oxidation reactions of amino-reduced ketones. Pyrazines accounted for a small percentage of the total odor-active compounds (about 2% to 3%). However, pyrazines were essential for the odor of SS and some baked foods, because of their lower odor threshold (0.04–30 μg/kg). Among the five pyrazines sniffed, methylpyrazine, 2,6-dimethylpyrazine, and 2-ethyl-5-methylpyrazine showed roasted and nutty aromas, which were consistent with the previous reports [33]. Moreover, 2,6-dimethylpyrazine and 2,3,5-trimethylpyrazine had higher content with the concentrations of 1077–1426 μg/L and 555–741 μg/L, respectively; both had also been determined as the key odorants in Chinese SS [37], Korean acid hydrolyzed SS [31], and Thai SS [38]. The concentrations of 2,5-dimethylpyrazine, 2,6-dimethylpyrazine, and methylpyrazine were close in A3 and AH1 samples; the levels of 2,5-dimethylpyrazine and 2-ethyl-5-methylpyrazine were similar in A6 and AH2 samples.

#### 3.3.8. Aldehydes

Some aldehydes were thought to be generated by deamination and decarboxylation reactions of free amino acids under the action of microorganisms. Furthermore, some studies had also indicated that aldehyde formation was related to the Maillard reaction [39]. Only two aldehydes were identified in this study; they were methional and phenylacetaldehyde. Phenylacetaldehyde (747–2288 μg/L) had been determined as the key odorants in Japanese SS and imparted a sweet aroma [40]; and it was derived from valine. Although methional from the degradation of methionine had a low level (328 μg/L -589 μg/L), its strong cooked potato-like aroma could be sniffed easily because of its lower threshold (0.45 μg/kg); it could be converted to methionol. Some factors, such as fermentation temperature, time, periodicity, etc., influenced the formation of methional in the process of SS production.

### 3.4. Heatmap Analysis of Odor-Active Compounds in 10 SS

A heatmap was made according to the concentrations of each odor-active compound. A color code was devised based on the scale from red to blue with their concentrations of compounds decreasing from high to low, which made it possible to make distinctions among the samples. In Figure 3a, it could be seen that A1 and A2, A3 and A4 were classified into one category, respectively, and AH1 was close to A3 and A4. In Figure 3b, A6 and AH2 were clustered into a group, which was consistent with the E-nose and E-tongue results. Based on the heatmap results, it could be concluded that the concentration of odor-active compounds in A3 (SSAT) and A4 (SSAT) was similar to that of AH1 (SSAA); A6 (SSAT) was similar to AH2 (SSAA).

### 3.5. OAVs Analysis

Odor activity value (OAV) of each aroma-active odorants was calculated based on the ratio of the concentration of the compound to its threshold [41]. Usually, a compound with larger OAV manifests that contributes more to the overall odor profile. As shown in Table 4, among 35 odorants sniffed, a total of 22 compounds had OAVs ≥ 1, and they were determined as the key odorants. HEMF (OAV:27027), 3-methyl-1-butanol (OAV:1756), and methional (OAV:1073) showed the higher OAVs, which contribute strong caramel, malt, and cooked potato-like odors, respectively. Baek and Kim employed SPME-GC-O to analyze the volatiles in Korean SS and also reported methional and HEMF had the most potent aroma-active compounds [42]. In addition, OAVs of five odorants, namely, 2AP (rice), acetoin (creamy), phenylacetaldehyde (honey-like), guaiacol (smoky), and HEMF (caramel), were greater than 100; HEMF and HDMF had been reported with higher OAVs in SS [27]. Although the intense sour odor could be sniffed during GC-MS-O analysis, OAVs of acetic acid and 3-methylbutanoic acid were less than 1, which was attributed to their higher thresholds. It was noteworthy that the number of the key odorants did not change during storage.

In Figure 3c, it could be seen that A1 and A2 were classified into one category, and A3 and A4 into the other category; AH1 was close to A3 and A4. In Figure 3d, A6 and AH2 were clustered into a group. From heatmap results, it could be concluded that the key odorants in A3 (SSAT) and A4 (SSAT) were similar to those of AH1 (SSAA), A6 (SSAT) similar to AH2 (SSAA).

### 3.6. Composition and Content Analysis of Amino Acids

Protein metabolism and changes in amino acids played the important roles in the overall taste of SS. Generally speaking, taste was one of the important factors in identifying the quality of SS. Normally, the umami in SS was provided by amino acids such as glutamic acid and aspartic acid. Due to the addition of sodium glutamate as an umami agent in SS, the following analysis excluded the proportion of glutamic acid. As shown in Table 5, in the first group (A1–A4 and AH1), the total content of amino acids in A3 (53.0 mg/mL), A4 (53.2 mg/mL), and AH1 (53.4 mg/mL) were very close. Proteins might break down during SS storage at room temperature or after heating treatment, the degradation increased under heating, and so, the amino acid content in AH1 was slightly higher. In the second group (A5–A8 and AH2), the content of amino acids in A6 (54.7 mg/mL), A7 (53.2 mg/mL), and AH2 (51.7 mg/mL) were close. Aspartic acid possessed the highest concentration among all amino acids, and it had the function of enhancing the umami of SS. During storage, the total amount of amino acids exhibited a trend of rising firstly and then decreasing, except for valine, phenylalanine, histidine, and cystine. Some amino acids might take part in Maillard reaction during storage, resulting in a decrease in their contents.

To find the similarity of these samples, HCA was performed based on the data listed in Table 5, and the results obtained are shown in Figure 4a, b. It could be seen that AH1 and A3 were clustered together, as were AH2 and A6, manifesting that they had the most similar amino acid composition. From the results above, taste characteristics of AH1 (SSAA) were close to A3 (SSAT), and AH2 (SSAA) close to A6 (SSAT). The conclusion obtained was nearly consistent with the result of E-nose and E-tongue analysis.

## 4. Conclusions

In summary, this present study first provides a comprehensive determination of the flavor relation of SSAT and SSAA by E-nose, E-tongue, and SE–SAFE coupled with GC-MS-O and amino acid analyzer. The overall flavor profile of SSAT for three weeks was similar to that of SSAA for one week, and six weeks (SSAT) was similar to two weeks (SSAA). By quantitating the key odorants and amino acids in SS stored at different condition, it was found that their concentrations in SSAT for three weeks were close to those in SSAA for one week, and six weeks (SSAT) were similar to two weeks (SSAA). It was concluded that the flavor changes in SSAA for one week were equal to those in SSAT for three weeks. The results obtained provide a new idea for details on the flavor changes in SS during storage.

## Figures and Tables

**Figure 1 foods-12-02707-f001:**
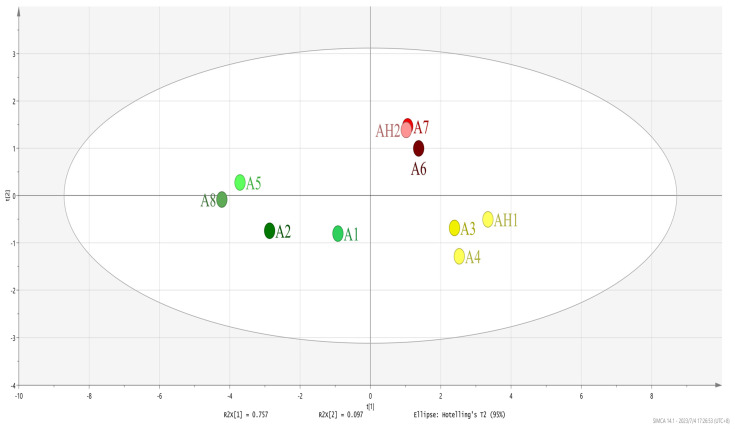
Overall odor profile of 10 soy sauce (SS) samples’ principal component analysis (PCA), obtained by electronic nose (E-nose).

**Figure 2 foods-12-02707-f002:**
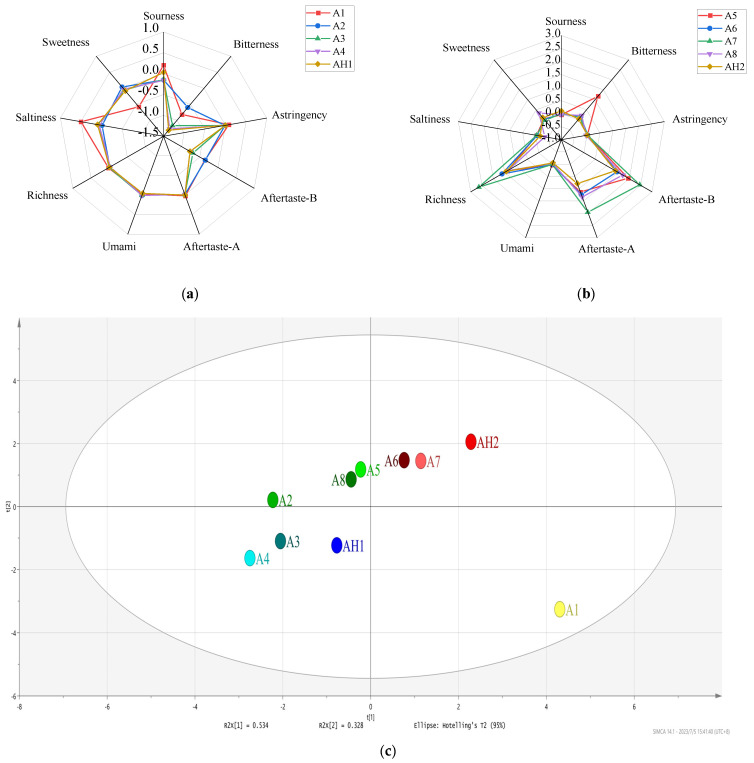
Overall taste profile of 10 soy sauce (SS) samples by electronic tongue (E-tongue): (**a**) A1–A4 and AH1 samples radar image of taste analyses; (**b**) A5–A8 and AH2 samples radar image of taste analyses; (**c**) A1–A8 and AH1–AH2 samples PCA of taste analyses.

**Figure 3 foods-12-02707-f003:**
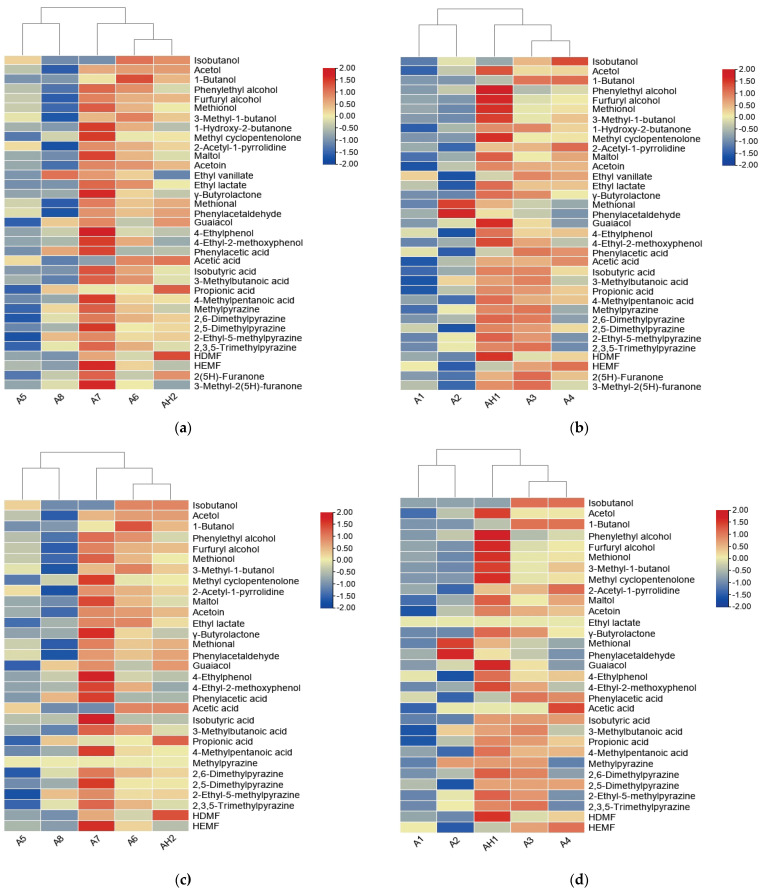
Aroma compound concentration heatmap of 10 soy sauce (SS) samples: (**a**) A1–A4 and AH1 samples heatmap; (**b**) A5–A8 and AH2 samples heatmap; odor activity value (OAV) heatmap of 10 soy sauce (SS) samples: (**c**) A1–A4 and AH1 samples; (**d**) A5–A8 and AH2 samples.

**Figure 4 foods-12-02707-f004:**
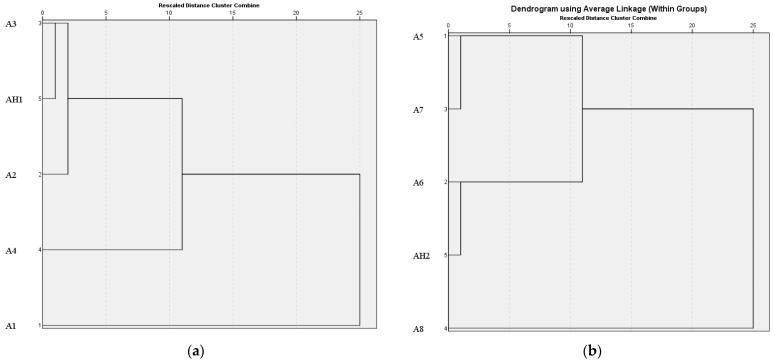
Amino acid hierarchical cluster analysis (HCA) of 10 investigated soy sauce (SS) samples: (**a**) A1–A4 and AH1 samples; (**b**) A5–A8 and AH2 samples.

**Table 1 foods-12-02707-t001:** Description of electronic nose sensor performance.

Nos.	Sensor Names	General Description
1	W1C	Aromatic components
2	W5S	Sensitive to nitrogen oxides
3	W3C	Ammonia, sensitive to aromatic compounds
4	W6S	Mainly hydrogen (selective)
5	W5C	Alkanes, aromatic constituents
6	W1S	Sensitive to methane
7	W1W	Sensitive to sulfur compounds
8	W2S	Sensitive to alcohol compounds
9	W2W	Aromatic components, sensitive to sulfur organic compounds
10	W3S	Sensitive to alkane components

**Table 2 foods-12-02707-t002:** Gas chromatography–mass spectrometry-olfactory (GC-MS-O) identified aroma-active compounds in soy sauce (SS) with the method of solvent extraction combined with solvent assisted flavor evaporation (SE-SAFE).

Nos. ^a^	Compounds	RI ^b^		CAS	Odor Description ^c^	Identification ^d^
		DB-WAX	HP-5MS			
1	Isobutanol	1099	ND ^e^	78-83-1	sour	O, MS, RI, S
2	1-Butanol	1149	ND	71-36-3	malty, balsam	O, MS, RI, S
3	3-Methyl-1-butanol	1213	745	123-51-3	malty	O, MS, RI, S
4	Methylpyrazine	1277	833	109-08-0	roasted, nutty	O, MS, RI, S
5	Acetoin	1294	723	513-86-0	creamy	O, MS, RI, S
6	Acetol	1309	704	116-09-6	sweet	O, MS, RI, S
7	2,5-Dimethylpyrazine	1335	917	123-32-0	roasted	O, MS, RI, S
8	2,6-Dimethylpyrazine	1341	917	108-50-9	roasted, nutty	O, MS, RI, S
9	2-Acetyl-1-pyrrolidine	1344	ND	85213-22-5	rice, popcorn	O, MS, RI
10	Ethyl lactate	1347	826	97-64-3	fruity, buttery	O, MS, RI, S
11	1-Hydroxy-2-butanone	1382	777	5077-67-8	sweet, coffee	O, MS, RI, S
12	2-Ethyl-5-methylpyrazine	1396	1003	13360-64-0	roasted, coffee	O, MS, RI, S
13	2,3,5-Trimethylpyrazine	1417	1007	14667-55-1	nutty, peanut	O, MS, RI, S
14	Acetic acid	1441	ND	64-19-7	sour	O, MS, RI, S
15	Methional	1461	912	3268-49-3	cooked potato-like	O, MS, RI, S
16	Propionic acid	1537	734	79-09-4	cheesy	O, MS, RI, S
17	Isobutyric acid	1569	ND	79-31-2	sour	O, MS, RI, S
18	Phenylacetaldehyde	1650	1050	122-78-1	honey, sweet	O, MS, RI, S
19	*γ*-Butyrolactone	1643	923	96-48-0	creamy, caramel	O, MS, RI, S
20	Furfuryl alcohol	1660	964	98-00-0	burnt	O, MS, RI, S
21	3-Methylbutanoic acid	1670	869	503-74-2	sweaty, cheese	O, MS, RI, S
22	Methionol	1724	985	505-10-2	cooked potato-like	O, MS, RI, S
23	3-Methyl-2(5*H*)-furanone	1729	ND	22122-36-7	roasted	O, MS, RI, S
24	2(5*H*)-Furanone	1766	ND	497-23-4	buttery	O, MS, RI, S
25	4-Methylpentanoic acid	1809	960	646-07-1	cheese	O, MS, RI, S
26	Methyl cyclopentenolone	1835	ND	80-71-7	caramel	O, MS, RI, S
27	Guaiacol	1865	1095	90-05-1	smoky	O, MS, RI, S
28	Phenylethyl alcohol	1919	1124	60-12-8	floral, rose	O, MS, RI, S
29	Maltol	1977	1128	118-71-8	sweet	O, MS, RI, S
30	4-ethyl-2-methoxyphenol	2034	1286	2785-89-9	smoky	O, MS, RI, S
31	HDMF	2038	1086	85554-61-6	caramel	O, MS, RI, S
32	HEMF	2072	ND	27538-09-6	caramel	O, MS, RI, S
33	4-Ethylphenol	2175	ND	123-07-9	smoky	O, MS, RI, S
34	Phenylacetic acid	2579	1263	103-82-2	honey	O, MS, RI, S
35	Ethyl vanillate	2633	1597	617-05-0	burnt	O, MS, RI, S

^a^ The aroma compounds identified on DB-Wax capillary column. ^b^ Retention index on capillaries DB-WAX and HP-5MS; RI did not exceed ± 50 of the library standard value. ^c^ Odor perception sensed at sniffing port. ^d^ O means identification by odor characteristic; MS means identification by comparison with the NIST 14 mass spectra database; RI means identification by retention index; S means confirmed by authentic standards; ^e^ ‘ND’ means the compound is not identified on the HP-5MS column.

**Table 3 foods-12-02707-t003:** The concentrations (μg/kg) and standard deviation of volatile compounds detected by gas chromatography–mass spectrometry-olfactory (GC-MS-O) in soy sauce (SS) samples.

Nos.	Compounds	*f* ^a^	Ions (*m*/*z*) ^b^	Concentrations (Mean ± Standard Deviation, μg/L) ^c^									
				A1	A2	A3	A4	AH1	A5	A6	A7	A8	AH2
1	Isobutanol	2.31	74	707 ± 6.73c	741 ± 21.3b	759 ± 6.27b	790 ± 5.47a	742 ± 5.39c	770 ± 130a	871 ± 20.3a	639 ± 51b	636 ± 18.6b	838 ± 14.5a
2	1-Butanol	0.68	56	5269 ± 109b	5268 ± 250b	5473 ± 99.1b	5479 ± 101b	5316 ± 436a	5198 ± 59.3d	6581 ± 20.5a	5783 ± 68c	5217 ± 35.5d	6008 ± 105b
3	3-Methyl-1-butanol	0.94	70	6596 ± 63.8c	6551 ± 184c	6832 ± 61b	6947 ± 43.7b	7276 ± 18.1a	7075 ± 1382a	8163 ± 159a	7625 ± 317a	5677 ± 130b	7480 ± 93.1a
6	Acetol	1.69	74	3556 ± 8.61d	3794 ± 74.1c	3905 ± 26b	3915 ± 24.8b	4168 ± 27.4a	4059 ± 530b	4562 ± 29.9a	4508 ± 176a	3594 ± 68.7c	4569 ± 60.7a
28	Phenylethyl alcohol	0.70	122	4550 ± 9.6b	4601 ± 48.4b	4586 ± 49.1b	4617 ± 38.2b	4802 ± 27.1a	4715 ± 836c	5533 ± 69.1ab	5776 ± 191a	4226 ± 15.9c	4871 ± 32.5bc
20	Furfuryl alcohol	1.03	98	4593 ± 13.1a	4562 ± 85.7a	4726 ± 35b	4765 ± 28.8b	5120 ± 13.6a	4896 ± 996b	5700 ± 55ab	6112 ± 162a	3931 ± 42.6c	5628 ± 59.1ab
22	Methionol	0.73	106	7062 ± 34.6c	6970 ± 158.3c	7242 ± 39.6b	7281 ± 28.3b	7688 ± 19.1a	7620 ± 1317b	8635 ± 57.9b	9689 ± 176a	6423 ± 25.3c	8039 ± 155b
11	1-Hydroxy-2-butanone	1.41	57	4657 ± 36.3d	4846 ± 25.7c	5134 ± 20.7a	5002 ± 24.7b	5121 ± 26.2a	5456 ± 511c	5920 ± 55.7ab	6304 ± 90.1a	5360 ± 32.9c	5525 ± 48.9bc
26	Methyl cyclopentenolone	0.67	112	485 ± 1.19c	485 ± 6.93c	502 ± 2.06b	504 ± 4b	532 ± 4.26a	511 ± 1.85e	591 ± 4.05c	710 ± 5.54a	575 ± 0.87d	598 ± 2.86b
9	2-Acetyl-1-pyrrolidine	1.00	83	67.0 ± 0.82c	58.0 ± 4.01d	84.0 ± 1.09b	96.0 ± 0.63a	83.0 ± 0.86b	73.7 ± 9.95c	83.3 ± 0.39b	91.5 ± 1.99a	40.4 ± 1.04d	75.7 ± 1.45bc
29	Maltol	0.95	126	7984 ± 50.5d	8080 ± 56.2cd	8181 ± 56.2bc	8270 ± 60.9ab	8363 ± 96.2a	8382 ± 1057c	9552 ± 109b	10,597 ± 258a	8066 ± 49.4c	8899 ± 71.7bc
5	Acetoin	1.70	88	1396 ± 8.08d	1484 ± 15.2c	1553 ± 12.4b	1547 ± 6.28b	1577 ± 11.5a	1607 ± 139c	1754 ± 4.85b	1766 ± 40.4b	1547 ± 16.8c	1722 ± 18.6b
35	Ethyl vanillate	0.78	91	127 ± 0.81c	77 ± 0.34e	148 ± 1.94a	140 ± 3.1b	111 ± 0.61d	156 ± 2.59d	178 ± 2.87c	187 ± 2.79b	194 ± 1.57a	153 ± 2.3d
10	Ethyl lactate	1.52	75	3054 ± 32.9b	2950 ± 69.7c	3123 ± 26.3ab	3123 ± 39.2ab	3192 ± 31.2a	2736 ± 31.3d	3754 ± 0b	3997 ± 34.4a	2767 ± 16.5d	3284 ± 34.7c
19	*γ*-Butyrolactone	1.40	86	1832 ± 5.88c	1826 ± 8.21c	1932 ± 9.62a	1888 ± 1.77b	1946 ± 18.5a	2054 ± 184c	2241 ± 10.7b	2526 ± 37.3a	2069 ± 17.5c	2128 ± 11bc
15	Methional	0.90	104	415 ± 9.37e	585 ± 12.4a	465 ± 16.4c	439 ± 14d	516 ± 8.05b	444 ± 66.9c	511 ± 13.5b	589 ± 26.6a	328 ± 8.53d	536 ± 15.3ab
18	Phenylacetaldehyde	0.27	91	1307 ± 9.04d	2288 ± 33.7a	1412 ± 17.3c	1216 ± 22.2e	1574 ± 43.1b	1197 ± 223b	1411 ± 24.7a	1589 ± 65.3a	747 ± 8c	1537 ± 99a
27	Guaiacol	0.60	124	211 ± 0.73d	218 ± 0.96c	221 ± 0.85b	211 ± 1.48d	237 ± 1.55a	244 ± 19.3c	264 ± 0.25b	287 ± 1.46a	276 ± 1.67ab	285 ± 1.37a
33	4-Ethylphenol	3.58	107	255 ± 1.34c	240 ± 2.52d	259 ± 1.82bc	260 ± 3.21b	268 ± 1.69a	267 ± 2.47e	307 ± 0b	493 ± 0.28a	300 ± 0.7c	292 ± 2.54d
30	4-ethyl-2-methoxyphenol	0.33	137	4512 ± 9.89d	4542 ± 28.8cd	4629 ± 19.4b	4559 ± 27.8c	4681 ± 13.5a	4934 ± 585c	5492 ± 20.2b	5892 ± 39.7b	5026 ± 45.5c	4967 ± 12.6c
34	Phenylacetic acid	0.62	91	2888 ± 45.6c	2167 ± 18.8e	3576 ± 56a	3421 ± 46.7b	2780 ± 85.8d	3693 ± 163d	3831 ± 63.8cd	4518 ± 72.3a	4219 ± 51.9b	3866 ± 31.3c
14	Acetic acid	1.36	60	8838 ± 55.5e	9576 ± 260d	10,286 ± 41.4c	10,507 ± 45.5b	10,335 ± 76.3a	10,782 ± 1796b	12,026 ± 161b	9321 ± 313c	8813 ± 140c	12,284 ± 200b
17	Isobutyric acid	1.18	73	522 ± 5.07d	544 ± 4.96c	595 ± 13.2a	571 ± 6.27b	597 ± 6.05a	638 ± 58.5b	685 ± 2.65ab	707 ± 5.52a	636 ± 9.59b	663 ± 9ab
21	3-Methylbutanoic acid	1.16	60	860 ± 6.37d	929 ± 4.18b	956 ± 3.99a	908 ± 7.89c	947 ± 3.79a	992 ± 134c	1113 ± 4.75ab	1155 ± 13.4a	949 ± 11.3c	1035 ± 3.86bc
16	Propionic acid	0.76	74	2249 ± 79.3c	2424 ± 94.2b	2603 ± 105a	2533 ± 106ab	2617 ± 93.2a	2661 ± 17.8c	2882 ± 73.9b	2874 ± 64.4b	2931 ± 54.3b	3071 ± 7.84a
25	4-Methylpentanoic acid	0.90	74	389 ± 0.76c	381 ± 1.28d	408 ± 2.03b	405 ± 3.61b	416 ± 1.71a	409 ± 3.44c	478 ± 1.09b	555 ± 3.62a	427 ± 46.4c	471 ± 5.21b
4	Methylpyrazine	0.44	94	430 ± 2.47d	454 ± 5.03b	473 ± 1.86a	443 ± 3.99c	470 ± 2a	458 ± 5.12e	550 ± 2.65b	600 ± 4.55a	540 ± 3.93c	515 ± 5.79d
8	2,6-Dimethylpyrazine	0.42	108	1077 ± 5.63b	1096 ± 62.9b	1166 ± 4.5a	1083 ± 15.7b	1175 ± 6.09a	1135 ± 16.4e	1366 ± 9.55b	1426 ± 7.43a	1292 ± 2.08d	1338 ± 16.85c
7	2,5-Dimethylpyrazine	1.07	108	337 ± 1.07ab	316 ± 30.3b	353 ± 1.65a	344 ± 1.5a	356 ± 2.67a	334 ± 2.67d	386 ± 32.4c	463 ± 1.28b	356 ± 5.64d	393 ± 1.13c
12	2-Ethyl-5-methylpyrazine	0.96	121	138 ± 0.68d	144 ± 1.35c	148 ± 0.21b	139 ± 1.84d	151 ± 0.46a	147 ± 2.22c	174 ± 4.17b	185 ± 0.86a	178 ± 1.88b	175 ± 1.88b
13	2,3,5-Trimethylpyrazine	0.37	122	555 ± 3.64c	573 ± 5.67b	592 ± 2.08a	556 ± 6.75c	588 ± 1.23a	574 ± 7.37d	698 ± 2.23b	741 ± 3.18a	659 ± 6c	653 ± 7.73c
31	HDMF	0.97	128	3485 ± 33.3e	3380 ± 25.9d	3641 ± 11.5c	3736 ± 26.7b	4118 ± 22.8a	3853 ± 405d	4256 ± 33.1c	4955 ± 99.8b	3708 ± 32.1d	5689 ± 77.6a
32	HEMF	1.00	142	28,756 ± 164bc	25,929 ± 105d	30,006 ± 125ab	30,807 ± 227a	33,389 ± 226c	31,348 ± 2681c	34,165 ± 65.3b	39,729 ± 382a	30,307 ± 214c	31,688 ± 257c
24	2(5H)-Furanone	0.99	84	559 ± 25.9bc	546 ± 17c	620 ± 26.6a	596 ± 26.4ab	601 ± 18.8ab	671 ± 34.9c	713 ± 46.8bc	786 ± 24.6a	713 ± 28.5bc	766 ± 40.2ab
23	3-Methyl-2(5*H*)-furanone	1.42	98	243 ± 1.36b	236 ± 3.16c	255 ± 0.71a	245 ± 0.94b	253 ± 1.57a	275 ± 18c	294 ± 1.87b	337 ± 3.18a	290 ± 1.95bc	275 ± 1.26c

^a^ Correction factor (f) by means of mixtures of known amounts of standard and internal standard. ^b^ The selected ions used for compound quantitation analysis. ^c^ The concentrations (μg/L) of volatile compounds quantified by GC-MS-O. Values with different superscript roman letters (a–e) in the same row are significantly different according to the Duncan test (*p* < 0.05).

**Table 4 foods-12-02707-t004:** Odor activity value (OAV) of volatile compounds detected in soy sauce (SS).

Nos.	Compounds	Threshold ^a^ (μg/kg)	OAVs ^b^									
			A1	A2	A3	A4	AH1	A5	A6	A7	A8	AH2
1	Isobutanol	6505	<1	<1	<1	<1	<1	<1	<1	<1	<1	<1
6	Acetol	10,000	<1	<1	<1	<1	<1	<1	<1	<1	<1	<1
2	1-Butanol	459	12	12	12	12	12	11	14	13	11	13
28	Phenylethyl alcohol	564	8	8	8	8	9	8	10	10	7	9
20	Furfuryl alcohol	4501	1	1	1	1	1	1	1	1	1	1
22	Methionol	123	57	57	59	59	62	62	70	79	52	65
3	3-Methyl-1-butanol	4	1649	1638	1708	1737	1819	1769	2041	1906	1419	1870
11	1-Hydroxy-2-butanone	- ^c^	-	-	-	-	-	-	-	-	-	-
26	Methyl cyclopentenolone	300	2	2	2	2	2	2	2	2	2	2
9	2-Acetyl-1-pyrrolidine	0.12	560	484	704	796	688	614	695	762	336	631
29	Maltol	1240	6	7	7	7	7	7	8	9	7	7
5	Acetoin	14	100	106	111	110	113	115	125	126	110	123
35	Ethyl vanillate	-	-	-	-	-	-	-	-	-	-	-
10	Ethyl lactate	50,000	<1	<1	<1	<1	<1	<1	<1	<1	<1	<1
19	γ-Butyrolactone	1000	2	2	2	2	2	2	2	3	2	2
15	Methional	0.45	923	1299	1033	977	1147	986	1136	1308	729	1192
18	Phenylacetaldehyde	6.3	208	363	224	193	250	190	224	252	119	244
27	Guaiacol	1.6	132	136	138	132	148	153	165	179	173	178
33	4-Ethylphenol	21	12	11	12	12	13	13	15	24	14	14
30	4-ethyl-2-methoxyphenol	89	51	51	52	51	53	55	62	66	56	56
34	Phenylacetic acid	12,000	<1	<1	<1	<1	<1	<1	<1	<1	<1	<1
14	Acetic acid	99,000	<1	<1	<1	<1	<1	<1	<1	<1	<1	<1
17	Isobutyric acid	6551	<1	<1	<1	<1	<1	<1	<1	<1	<1	<1
21	3-Methylbutanoic acid	490	2	2	2	2	2	2	2	2	2	2
16	Propionic acid	2190	1	1	1	1	1	1	1	1	1	1
25	4-Methylpentanoic acid	810	<1	<1	<1	<1	<1	<1	<1	<1	<1	<1
4	Methylpyrazine	30,000	<1	<1	<1	<1	<1	<1	<1	<1	<1	<1
8	2,6-Dimethylpyrazine	718	2	2	2	2	2	2	2	2	2	2
7	2,5-Dimethylpyrazine	1750	<1	<1	<1	<1	<1	<1	<1	<1	<1	<1
12	2-Ethyl-5-methylpyrazine	40	3	4	4	3	4	4	4	5	4	4
13	2,3,5-Trimethylpyrazine	350	2	2	2	2	2	2	2	2	2	2
31	HDMF	22	156	152	163	168	185	173	191	222	166	255
32	HEMF	1.15	250,06	22,547	26,092	26,789	24,413	27,259	29,709	34,547	26,354	27,555
24	2(5*H*)-Furanone	714	<1	<1	<1	<1	<1	<1	<1	<1	<1	<1
23	3-Methyl-2(5*H*)-furanone	-	-	-	-	-	-	-	-	-	-	-

^a^ The threshold of volatile compounds in water referred in the literature. [41,43]; ^b^ The odor activity values of volatile compounds in SS. ^c^ “-” means not detected.

**Table 5 foods-12-02707-t005:** Amino acids concentrations of soy sauce (SS) processed by different storage conditions.

Amino Acid	Concentrations (mg/mL) ^a^									
	A1	A2	A3	A4	AH1	A5	A6	A7	A8	AH2
Aspartic acid	7.77	7.90	8.03	8.26	8.09	7.85	8.27	7.95	7.53	7.98
Glycine	3.11	3.16	3.20	3.26	3.22	3.09	3.29	3.17	3.00	3.15
Alanine	3.53	3.52	3.63	3.64	3.65	3.28	3.76	3.59	3.39	3.58
Threonine	2.67	2.64	2.75	2.77	2.79	2.70	2.81	2.72	2.58	2.71
Serine	3.59	3.71	3.73	3.88	3.76	3.64	3.84	3.66	3.50	3.74
Proline	4.03	4.06	4.18	4.30	4.24	4.12	4.32	4.14	3.91	4.15
Valine	4.49	4.06	4.49	4.13	4.55	4.27	4.60	4.67	4.27	4.15
Isoleucine	3.56	3.59	3.65	3.70	3.68	3.55	3.76	3.64	3.43	3.61
Leucine	5.24	5.27	5.39	5.49	5.45	5.28	5.56	5.37	5.08	5.33
Phenylalanine	3.81	3.38	3.79	3.57	3.79	3.76	3.94	3.96	3.51	3.53
Histidine	1.67	1.64	1.67	1.71	1.62	1.65	1.71	1.67	1.57	1.61
Tyrosine	0.87	0.91	0.90	0.94	0.91	0.88	0.94	0.89	0.85	0.90
Lysine	3.98	3.99	4.09	4.19	4.13	4.04	4.21	4.05	3.83	4.05
Arginine	1.40	1.40	1.44	1.47	1.43	1.42	1.47	1.43	1.35	1.38
Cystine	1.09	0.85	1.02	0.78	1.06	0.05	1.11	1.27	0.98	0.80
Methionine	1.05	1.26	1.05	1.12	1.03	0.89	1.07	1.06	0.98	1.01
	51.8	51.3	53.0	53.2	53.4	50.5	54.7	53.2	49.8	51.7

^a^ The concentrations (mg/mL) of amino acids quantified.

## Data Availability

The datasets generated for this study are available on request to the corresponding author.

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
