# Peer review of "Comparison of Flavor Profile Relationship of Soy Sauce under Different Storage Conditions"

_foods, 2023, doi:10.3390/foods12142707_

Round 1

Reviewer 1 Report

The manuscript is interesting and describes a new aspect of fermented food. I want to propose some improvements:

Line 44 – Please change duck into duck meat products or something like that. Just duck could suggest living animals.

The introduction part seems to need more about soy sauce production. For readers not familiar with the process production, the fact in text in line 278 - under aerobic conditions, and some alcohols were also formed by yeast; line 324 by Aspergillus oryzae and Aspergillus niger could be completely not clear.

 Line 312-314 – Moreover, concentration of both compounds in AH1 was significantly higher than A1-  A4 samples, which implied that heating could increase the contents of them. Could you please explain why?

Reviewer 2 Report

The following article titled “Comparison of flavor profile relationship of soy sauce under different
storage conditions” studied flavor profile of soy source with two different storage condition at 25 and 36 as an accelerated condition using by e-nose, e-tongue, GC-MS-O.

The research has an interesting point regarding comparison of flavor profiles of two different temperature. However, the article contained many big concerns. The detail addressed as follow;

1.     You have to explain why you choose two temperature 25oC and 36 oC. Actually the 36 oC is normal temperature at summer season so, it is hard to say 36 is an accelerating temperature. As an accelerating condition, I suggest above 40 oC

2.     This is biggest concern that Figure 1 seems generated separately like as A0~A4 and AH1 as a one group. A5~A8 and AH2 analyzed as another group. This is wrong way. The PCA analysis analyzed all sample group as at a time.

3.     How many time experiments carried out for A0~A8, AH1 and AH2 sample. You need to put the replicate number.

Minor things. Please modify them

1.     The format is not following ‘Foods’

2.     Abstract contained unnecessary ‘-‘

3.     The reference is not consistency and not followed ‘Foods’ format.

The following article titled “Comparison of flavor profile relationship of soy sauce under different
storage conditions” studied flavor profile of soy source with two different storage condition at 25 and 36 as an accelerated condition using by e-nose, e-tongue, GC-MS-O.

The research has an interesting point regarding comparison of flavor profiles of two different temperature. However, the article contained many big concerns. The detail addressed as follow;

1.     You have to explain why you choose two temperature 25oC and 36 oC. Actually the 36 oC is normal temperature at summer season so, it is hard to say 36 is an accelerating temperature. As an accelerating condition, I suggest above 40 oC

2.     This is biggest concern that Figure 1 seems generated separately like as A0~A4 and AH1 as a one group. A5~A8 and AH2 analyzed as another group. This is wrong way. The PCA analysis analyzed all sample group as at a time.

3.     How many time experiments carried out for A0~A8, AH1 and AH2 sample. You need to put the replicate number.

Minor things. Please modify them

1.     The format is not following ‘Foods’

2.     Abstract contained unnecessary ‘-‘

3.     The reference is not consistency and not followed ‘Foods’ format.

Reviewer 3 Report

The results in this manuscript are solid. However, it requires some revisions.

1. Please follow the format of Foods, MDPI. Figures and tables should be integrated within main text.

2. Page 1, line 9 and line 11:  ‘-’ should be removed from ‘ana-lytical’ and ‘sol-vent’.

3. Page 1, lines 38-40: there are few publications…. Please add those publications as references.

4. Page 3, lines 104-105: Please add city and country names after these two companies.

5. Page 3, line 126: Please add city name before Japan.

6. Page 3, line 142: Extracting references from previous studies. This sentence should be rewrite.

7. Page 3, lines 144-147: Please add city and country names after these two companies.

8. Page 4, line 206: Why not use M to replace mol/L?

9. Page 5, line 216: Please provide more details for origin 2021b.

10. Page 5, lines 249 and 256: Shu et al., 2015 and Kim et al., 2015 are wrong ref. format.

11. Page 7, lines 363-366: It was reported ….of raw SS. Please provide the reference.

12. Page 10, line 488: ‘Author Contributions’ is mandatory.

13. Some references are incomplete, for example ref. 18, 20, 26, 27, 40, and 41. Page numbers are missing.

14. Table 5: column A1: Umami total 7.77 + Sweetness total 16.9 + Bitterness total 25.0 + Tasteless total 2.14 = 51.81. However, the last total is 51.9.

15. Figure 2, (I) in the middle should be removed, and the right (II) should be moved to the central.  

Minor editing of English language required

Round 2

Reviewer 2 Report

The article has been modified by followed review's comments.  So, it is acceptable to 'Foods'. However, abbreviation of Journal at references needs to correction one more time.  Some abbreviation has a comma some has not. 

English language looks fine for me. 

Reviewer 3 Report

This revised manuscript is improved a lot. To me, it is acceptable.